# Pt(IV) Prodrugs with NSAIDs as Axial Ligands

**DOI:** 10.3390/ijms22083817

**Published:** 2021-04-07

**Authors:** Daniil Spector, Olga Krasnovskaya, Kirill Pavlov, Alexander Erofeev, Peter Gorelkin, Elena Beloglazkina, Alexander Majouga

**Affiliations:** 1Department of Materials Science of Semiconductors and Dielectrics, National University of Science and Technology (MISIS), Leninskiy Prospect 4, 101000 Moscow, Russia; erofeev@polly.phys.msu.ru (A.E.); peter.gorelkin@gmail.com (P.G.); alexander.majouga@gmail.com (A.M.); 2Chemistry Department, Lomonosov Moscow State University, Leninskie Gory 1,3, 119991 Moscow, Russia; kirill.pavlov2011@mail.ru (K.P.); beloglazki@mail.ru (E.B.); 3Mendeleev University of Chemical Technology of Russia, Miusskaya Ploshchad’ 9, 125047 Moscow, Russia

**Keywords:** platinum, prodrugs, NSAIDs

## Abstract

A chemo-anti-inflammatory strategy is of interest for the treatment of aggressive cancers. The platinum (IV) prodrug with non-steroidal anti-inflammatory drugs (NSAIDs) as axial ligands is designed to efficiently enter tumor cells due to high lipophilicity and release the cytotoxic metabolite and NSAID intracellularly, thereby reducing side effects and increasing the therapeutic efficacy of platinum chemotherapy. Over the last 7 years, a number of publications have been devoted to the design of such Pt(IV) prodrugs in combination with anti-inflammatory chemotherapy, with high therapeutic efficacy in vitro and In vivo. In this review, we summarize the studies devoted to the development of Pt(IV) prodrugs with NSAIDs as axial ligands, the study of the mechanism of their cytotoxic action and anti-inflammatory activity, the structure–activity ratio, and therapeutic efficacy.

## 1. Introduction

Platinum-containing antineoplastic agents are effective anticancer drugs that are used in 50% of all chemotherapy regimens in clinical practice [1]. Three FDA-approved platinum-containing anticancer drugs have been applied around the world for decades: cisplatin, carboplatin, and oxaliplatin [2]. Also, a second generation drug, nedaplatin, is approved in Japan for the treatment of lung, esophageal, and head and neck cancers; lobaplatin is approved in China for the treatment of inoperable metastatic breast cancer and small-cell lung cancer [3]; and heptaplatin is approved in Korea for the treatment of stomach cancer [4].

Pt(II) coordination compounds easily undergo non-selective ligand substitution on the way to the tumor site, while approximately 90% of the administered cisplatin is deactivated in the bloodstream due to irreversible binding to albumin and other plasma proteins, with only 1% (or less) binding with the intended target (nuclear DNA) [5]. Protein binding to cisplatin leads to the formation of toxic metabolites that cause serious side effects such as kidney damage, nerve damage, and hearing loss [6]. Cisplatin was also reported to cause severe delayed emesis after 24 hours of administration [7]. The second-generation platinum anticancer drugs, carboplatin and oxaliplatin, are less toxic than cisplatin but show no improvement in potency or selectivity [8,9]. Various strategies have been suggested to improve the efficiency of platinum drugs [10]. One of the approaches to overcome these and the other well-known disadvantages of Pt(II)-based drugs, such as low bioavailability and rapidly emerging resistance, is to use Pt(IV) complexes as prodrugs. Pt(IV) have a low-spin octahedral d_6_ geometry and have the potential to overcome the disadvantages associated with platinum-containing chemotherapeutic agents [11]. The oxidation state of Pt(IV) provides an increased inertness of the coordination compounds compared to the Pt(II) complexes, and the conjugation of additional ligands in the axial position allows for both adjusting the lipophilicity and increasing the selectivity of the coordination compounds to tumor cells [12].

Chronic inflammation is an important therapeutic target in both the treatment of malignant neoplasms and drug development [13]. Various isoforms of cyclooxygenase (COX), such as COX-1 and COX-2, are markers of chronic inflammation, catalyzing the rate-limiting stage of prostaglandin formation, which plays a key role in the formation of the immune response [14]. Since COX-2 and prostaglandin-2 are involved in the inflammatory response, the suppression of apoptosis, and the formation of drug resistance, the inhibition of COX-2 can reduce inflammation and, as a result, reduce the rate of metastasis of malignant neoplasms [15]. Drug resistance is also associated with epithelial–mesenchymal transition (EMT), which is the transdifferentiation of epithelial cells into motile mesenchymal cells [16]. EMT-activated cells release matrix metalloproteinases (MMPs); the increased expression of MMPs contributes to the development of malignant neoplasms [17]. In addition, EMT transition is accompanied by the overexpression of COX-2, which can lead to immune evasion and tumor drug resistance [18].

Non-steroidal anti-inflammatory drugs (NSAIDs) are capable of inhibiting various isoforms of cyclooxygenase, which makes them a promising addition to existing chemotherapy regimens, both in view of their anti-inflammatory and analgesic effects, but also their ability to reduce the rate of metastasis and mount an immune response [19]. The development of Pt(IV) prodrugs with COX inhibitors in the axial position is a promising therapeutic approach due to the fact that the Pt(IV) prodrug’s ability to bind COX-2 will increase the selectivity of prodrugs for tumor cells in comparison with classical Pt(II) drugs. Also, an increase in the lipophilicity of the Pt(IV) prodrug compared to the initial Pt(II) drugs significantly increases drug intracellular accumulation, thereby increasing its effectiveness and reducing side effects due to a decrease in the effective dose. Thus, the design and synthesis of Pt(IV) prodrugs modified with NSAID moieties will make it possible to obtain effective drugs with increased selectivity for tumor cells, higher efficacy compared to classical platinum-containing drugs, as well as an additional anti-inflammatory effect that provides a synergistic effect with chemotherapy and reduces side effects.

Since 2014, after the report of the cisplatin-based Pt(IV) prodrug with aspirin as the axial ligand, asplatin, which is capable of simultaneously releasing the NSAID aspirin and cisplatin by Pathak et al. [20], lots of publications have been devoted to the design of Pt(IV) prodrugs based on different platinum(II) drugs with non-steroidal anti-inflammatory drugs as axial ligands (Figure 1), and most of the prodrugs developed so far have demonstrated high therapeutic efficiency in vitro and In vivo. In this review, we have summarized the results obtained over the years, and examined the data obtained, the mechanisms of cytotoxic action, and the data from In vivo studies.

In the first part of this review, we summarize dual-action Pt(IV) prodrugs with one or two NSAID moieties as axial ligand(s) (Table 1). In the second part of this review, we summarize unsymmetrical triple action Pt(IV) prodrugs with different bioactive axial ligands, including a study of the mechanism of their cytotoxic action, anti-inflammatory activity, the structure–activity ratio, and therapeutic efficacy.

## 2. Dual-Action Pt(IV) Prodrugs with NSAID Axial Ligands

### 2.1. Aspirin

In 2014, Pathak et al. designed a conjugate of one aspirin moiety with a platinum core named asplatin (or Platin-A) 1, which was the first reported Pt(IV) prodrug with an NSAID as an axial ligand [20] (Figure 2). Acetylsalicylic acid (aspirin) is able to bind irreversibly with COX-1 and COX-2 and reduce the severity of cisplatin therapy side effects such as ototoxicity and nephrotoxicity [30,31]. The synthesis of asplatin **1** was performed by the acylation of oxoplatin by aspirin anhydride.

The ability of asplatin **1** to reduce was measured using cyclic voltammetry (CV) under two pH values. The change of pH value from 7.5 to a more acidic value of 6.4 was followed by E(Pt^+4^/Pt^+2^) reduction potential increase by 42 mV, thus allowing the authors to assume that prodrug **1** will be easily reduced in tumor tissues. In model conditions that mimic intracellular reduction, Platin-A **1** was incubated with sodium ascorbate and 2′-deoxyguanosine 5′-monophosphate sodium salt hydrate (5′-GMP). Analysis of the reaction products by matrix-assisted laser desorption ionization (MALDI) mass-spectrometry (MS) showed the formation of Pt-GMP_2_ bis adduct, [Pt(NH_3_)_2_(5′-GMP)_2_], thus proving the ability of Platin-A **1** reduction products to interact with guanosine bases in DNA [20].

The release of aspirin upon reduction of the prodrug was proved by HPLC; upon the reduction of prodrug **1** in the presence of sodium ascorbate, an increasing amount of aspirin and salicylic acid release from the prodrug was detected. This proves that platinum (IV) prodrugs are able to release active platinum(II) moieties and axial ligands in either pristine or metabolite form upon reduction [20].

A cytotoxicity assay demonstrated that the antiproliferative ability of asplatin was comparable to that of cisplatin (Table 2).

Asplatin **1** showed a similar ability to cisplatin to induce early apoptosis in tumor cells. Moreover, the ability of prodrug **1** to inhibit COX-1 and COX-2 was close to that of aspirin, which proves that prodrug **1** possesses both cytotoxic and anti-inflammatory properties. Asplatin **1** also suppressed tumor necrosis factor-α (TNF-α) and interleukin (IL)-6, which causes an inflammatory response, while at the same time induced the secretion of the anti-inflammatory cytokine IL-10. Thus, the conjugation of the NSAID aspirin with cisplatin resulted in a potent dual-action chemo-anti-inflammatory Pt(IV) prodrug capable of the simultaneous release of both anti-inflammatory and anticancer drugs, along with increased efficacy and reduced side effects, as compared to classical platinum chemotherapy [20].

### 2.2. Ibuprofen

Neumann et al. designed two Pt(IV) prodrugs with two moieties of indomethacin or ibuprofen in the axial position [21] (Figure 3). Prodrugs **2** and **3** were synthesized by reacting the corresponding acyl chloride with oxoplatin in acetone in the presence of pyridine.

The cyclic voltammetry assay showed a great difference between the reduction potentials of prodrugs **2** and **3** (−0.36 V and −0.68 V, respectively), which Neumann et al. suggested is probably due to the strong electron-withdrawing indole system and increased steric bulk of indomethacin. Thus, a weakened bond could probably facilitate ligand dissociation, resulting in poor stability. The stability of prodrugs **2** and **3** in a reducing environment was studied in the presence of ascorbic acid over three days. The release of axial ligands was detected by ^1^H NMR. The release of ibuprofen and prodrug decay was observed as additional peaks in ^1^H NMR and appeared in the –CH_3_ group region at 1.38 ppm and the –CH_2_– group region at 2.9 ppm. Approximately 40% of each drug was reduced during the experiment, which was evaluated by the authors as comparable with the clearance rate of platinum-based drugs from the body [21].

The COX-inhibiting ability of prodrugs **2** and **3** towards purified COX-1 and COX-2 isoforms demonstrated that complex **2** was a highly selective inhibitor of COX-2, while complex **3** did not demonstrate any significant inhibiting activity on either isoform. 

The antiproliferative activity of prodrugs **2** and **3** with cisplatin as a control was assessed on the colon cancer cell line HCT-116, the ovarian cancer cell line OVCAR3, the triple negative breast cancer cell line MDA-MB-231, and the head and neck squamous cell carcinoma cell line 1483 HNSCC via (3–(4,5–dimethylthiazol–2–yl)–2,5–diphenyltetrazolium bromide (MTT) assay (Table 3). Prodrugs **2** and **3** both showed antiproliferative activity surpassing that of cispaltin. It is also worth noting the extremely high activity of **3** towards MDA-MB-231 cells (cisplatin-resistant triple negative breast tumor cell line), where it was about 400-fold more toxic than cisplatin [21].

Neumann et al. also designed oxaliplatin-based prodrugs **4** and **5** with indomethacin and ibuprofen as axial ligands [22]. Prodrugs **4** and **5** were obtained following the procedure utilized for synthesis of coordination compounds **2** and **3**, by the reaction of oxaliplatin with the corresponding acid chloride [21] (Figure 4)**.**

Cyclic voltammetry was used to evaluate how easily prodrugs **4** and **5** could be reduced, since both too high and too low reduction potentials are detrimental to the activity of the complexes [32,33]. The reduction potentials obtained by CV experiments were compared to those obtained for cispaltin-based prodrugs **2** and **3** [21]. Contrary to the reduction potentials of prodrugs **2** and **3**, in the case of prodrugs **4** and **5**, no significant difference in reduction potential was observed (−0.52 V and −0.56 V, respectively).

The ability of prodrugs **4** and **5** to inhibit cyclooxygenases was studied on purified isoforms of ovine COX-1 and murine COX-2. Neither of the isoforms were inhibited by ibuprofen-containing complexes **3** or **5**. However, while prodrug **2** with indomethacin in the axial position acts as a selective COX-2 inhibitor, its oxaliplatin analogue **4** demonstrated selectivity towards COX-1 and no activity against COX-2. Docking studies for prodrugs **2** and **4** were carried out to gain insight into the differences in binding of prodrugs **2** and **4** to COX-2. The results suggest that compound **4** cannot bind to COX-2 due to the constricted entrance in the COX-2 active site, which prevents the bulky oxaliplatin equatorial ligands of **4** from accessing the enzyme. The high inhibitory activity of prodrug **2** towards the COX-2 isoform is attributed to the additional activity of the second indomethacin moiety, while only one moiety of indomethacin in prodrug **4** interacts with the enzyme active site [22].

The cytotoxicity of complexes **4** and **5** was evaluated by MTT assay on HCT 116 (colorectal adenocarcinoma) and MDA-MB-231 (triple negative breast adenocarcinoma) cells (Table 4). Both prodrugs **4** and **5** showed sub-micromolar activity towards MDA-MB-23 cells, while prodrug **5** demonstrated much lower IC_50_ values than prodrug **4** on HCT-116 cells (0.31 µM and 1.3 µM, respectively). However, both complexes **4** and **5** were less potent than cisplatin-based prodrug **3** with ibuprofen axial ligands. Cellular accumulation of prodrugs **2–5** was studied via ICP-MS on HCT-116 and MDA-MB-231 cell lines (Table 5). Interestingly, no direct correlation between cytotoxicity and intracellular accumulation of prodrugs **2–5** was found, as intracellular platinum accumulation in both HCT-116 and MDA-MB-231 cells for the most potent drug (**3**) was lower than for its less potent oxaliplatin analogue (**5**) and the cisplatin–indomethacin complex (**2**). The trend in platinum accumulation cannot be solely attributed to the lipophilicity of prodrugs **2–5**; indomethacin is more lipophilic than ibuprofen (logP values are 4.27 and 3.5, respectively), however, no correlation between axial ligands and cellular uptake of prodrugs **2–5** was observed [22].

Curci et al. designed kiteplatin-based Pt(IV) prodrug **6** with two ibuprofen moieties as axial ligands [23]. The prodrug was obtained by the acylation of oxidized kiteplatin by the acid chloride of ibuprofen (Figure 5). The structure of the resulting complex was confirmed by thorough analysis of multiple NMR experiments, including 2D COSY spectrum and [^1^H-^13^C]-HSQC 2D NMR. ^195^Pt NMR showed the presence a single peak, confirming the formation of only one platinum complex.

To determine the stability of prodrug **6**, the CV electrochemical curves were recorded on a glassy carbon electrode. The determined reduction potential was found to be −0.93 V and is compatible to the potentials of similar platinum(IV) coordination compounds [34]. The cytotoxicity of the compound **6** was studied on non-COX-expressing colorectal carcinoma cell lines. Prodrug **6** showed submicromolar IC_50_ values, which was up to 42-fold higher than platinum(II) drugs in the assay (Table 6).

### 2.3. Flurbiprofen

Tan et al. designed a cisplatin-based prodrug **7** with two flurbiprofen moieties in axial positions [24]. Synthesis of the compound was performed by the reaction of oxoplatin with flurbiprofen acid chloride in tetrahydrofuran (THF) (Figure 6). The formation of the complex was confirmed by observing the shift of the –CH_3_ methyl group from 1.41 ppm in free flurbiprofen to 1.38 ppm in the ^1^H NMR spectrum of prodrug **7**. The reduction potential obtained by CV was −0.68 V at pH 7.4, which allowed the authors to assume that prodrug **7** is capable of reduction in a cellular environment.

The stability of prodrug **7** in the reducing environment was assessed in the presence of ascorbic acid as a reductant and showed complete decay of the complex within 48 hours, as determined by HPLC. To establish whether the Pt(IV) prodrug can release the Pt(II) DNA-binding moiety, compound **7** was incubated with ascorbic acid in the presence of 2′-deoxyguanosine 5′-monophosphate sodium salt hydrate (5-GMP), a nucleotide base of DNA. Analysis of the products by ESI-MS showed the formation of bifunctional Pt(II)/GMP adduct [Pt(NH_3_)_2_(5′-GMP)_2_]^+^ with the observed *m*/*z* of 922.46. Thus, platinum(IV) prodrug **7** is capable of releasing cisplatin in reducing environments, which then binds to DNA [34].

The cytotoxicity of prodrug **7** was evaluated by MTT assay (Table 7). The IC_50_ values of complex **7** were up to 82-fold lower than for cisplatin and the equimolar mixture of cisplatin and flurbiprofen. It is worth noting that prodrug **7** showed activity against the cisplatin-resistant cell line A549-DDP. Complex **7** was equally active on both cisplatin-sensitive and cisplatin-resistant A-549 cells (2.7 µM and 2.5 µM, respectively), while cisplatin showed an almost 3-fold decrease in activity (7.4 µM for A549 and 20.03 µM for A549-DDP). Cellular platinum accumulation was determined in BEL7404, BEL7404-CP20, and SW480 cells after 6 hours of incubation with 10 µM of prodrug **7**, cisplatin, and an equimolar mixture of cisplatin and flurbiprofen. Platinum content in cells incubated with prodrug **7** was 21- to 57-fold more than with cisplatin; DNA platination was 5- to 11-fold more for prodrug **7** than for cisplatin [23].

An opalescence and the Tyndall effect of prodrug **7** stock solution was observed; thus, dynamic light scattering (DLS) analysis was conducted which showed that in solution, prodrug **7** formed nanoparticles with a diameter of 115.3 nm. The formation of nanoparticles was further proved by scanning electron microscopy (SEM), transition electrone microscopy (TEM) and atomic force microscopy (AFM). The obtained results led the authors to the assumption that the difference in cellular uptake between prodrug **7** and cisplatin is due to nanoparticle formation [23].

### 2.4. Naproxen

Ravera et al. synthesized platinum(IV) prodrugs **8** and **9** based on cisplatin with two NSAIDs, naproxen and ketoprofen, in the axial positions (Figure 7) [25]. As a reference compound, asplatin **1** was chosen and synthesized as well. Asplatin **1** was prepared as described by Pathak et al. [20]. Prodrugs **8** and **9** were prepared by the acylation of [Pt(NH_3_)_2_(Cl)_2_(OH)(OAc)] by the corresponding acid chloride.

The lipophilicities of all three Pt(IV) prodrugs **1**, **8** and **9** were evaluated using RP-HPLC (Table 8). The logarithm of the RP-HPLC capacity factor k’ usually correlates with the octanol/water partition coefficient [35]. The obtained log k’ values for prodrugs **1**, **8** and **9** follows the same trend as for their corresponding NSAIDs. Asplatin **1** was found to be the least lipophilic, while **9** was the most lipophilic coordinating compound.

The cytotoxicity of compounds **1, 8** and **9** was evaluated via MTT assay using both cyclooxygenase (COX)-expressing and non-COX-expressing cell lines (Table 8).

The results of the MTT assay demonstrated the significant antiproliferative activity of coordination compounds **8** and **9**, which were up to 13-fold more toxic than asplatin **1** and up to 20-fold more toxic than cisplatin. No correlation between COX-2 expression and cytotoxicity was found. Based on the results of the MTT assay, lipophilicity can be assumed to be the main factor for prodrug cytotoxic activity, as the most lipophilic complex **9** is the most toxic, while the least lipophilic compound (cisplatin) has the highest IC_50_ values on nearly all cell lines [25].

In order to confirm the relationship between cytotoxicity and lipophilicity, the accumulation ratio (AR) was evaluated as the ratio between intra- and extracellular platinum concentration. Cells were incubated with 10 µM of cisplatin and prodrugs **1**, **8** and **9** for 4 h. The most lipophilic prodrugs **8** and **9** demonstrated higher ARs than the less lipophilic cisplatin and asplatin **1** with AR values of 14. AR values of platinum(IV) prodrugs **8** and **9** were 14 and 12, respectively, while cisplatin and Platin-A **1** demonstrated AR values of 2 and 1, respectively. The presence of lipophilic moieties in axial positions enhances the ability to penetrate the cellular membrane, leading to increased platinum uptake and, consequently, antiproliferative activity [36].

To confirm the assumption that the mechanism of the toxicity of prodrugs **8** and **9** is COX-independent, a RT-qPCR analysis was carried out. Two cell lines, one with high COX-2 expression (A-549) and one with low COX-2 expression (HCT-116), were chosen. Five genes were chosen to study the influence of prodrugs **8**, **9**, cisplatin, naproxen, and ketoprofen: Bcl-2 family genes that regulate apoptosis, COX-2, and NSAID-activated gene NAG-1. The pro-apoptotic proteins BAD (BCL2 associated agonist of cell death) and BAX (Bcl-2-associated X protein) were upregulated in the presence of all three platinum compounds in both cell lines, while the anti-apoptotic gene BCL-2 was downregulated. There is evidence that BAX upregulation promotes apoptotic activity through caspase activation [37]. COX-2 expression was upregulated by all compounds (**1**, **8**, **9**, cisplatin, naproxen and ketoprofen) on the low COX-expressing cells HCT-116. The NAG-1 gene, which is involved in antiproliferative activity [38], was expressed by prodrugs **8**, **9** and cisplatin on both cell lines, including HCT-116, in which little to no COX-2 activity was observed.

To sum up, the antiproliferative activity of platinum(IV) prodrugs **8** and **9** was shown to occur through a COX-independent pathway, with lipophilicity being the key factor determining the efficiency of these compounds.

In another study devoted to the synthesis of Pt(IV) prodrugs with naproxen in axial positions, Tolan et al. designed six platinum–naproxen prodrugs derived from three Food and Drug Administration (FDA)-approved platinum(II) drugs [26] (Figure 8). Prodrugs **10–12** with a cisplatin, carboplatin and oxaliplatin core, respectively, were obtained by the reaction of oxoplatin with the N-hydroxysuccinimide (NHS)-ester of naproxen. Platinum prodrugs **13–16** with benzoic acid, succinic acid, glutaric acid and moieties, respectively, were synthesized by the reaction of complex **10** with the anhydride of the corresponding acid. Such an approach to the design of unsymmetrical platinum(IV) prodrugs allowed these authors to finely tune the properties of the resulting complex [39]. The second axial ligand of the platinum(IV) prodrugs **13–15** was varied to alter the resulting lipophilicities of the platinum(IV) prodrugs.

The antiproliferative activity of prodrugs **10–15** was evaluated by MTT assay on the breast cancer cell line MCF-7 and the triple negative breast adenocarcinoma cell line MDA-MB-231 (Table 8). The cytotoxicity of all prodrugs on the MCF-7 cell line was 1.5- to 2-fold higher than that of cisplatin. The results of the MTT assay indicate an 11- to 30-fold difference between the IC_50_ values of cisplatin and compounds **10–15**. It is worth noting that prodrug **13**, with the most lipophilic benzoate ligand in the axial position, showed the lowest IC_50_ values among all prodrugs tested (**10–15**) [26].

Cell death mode was assessed by fluorescent staining of MCF-7 and MDA-MB-231 cells incubated with the most active Pt(IV) prodrug **13**. The results obtained indicate that prodrug **13** induces necrosis and apoptosis (early and late, combined) of 30.56% and 22.68% cells, respectively, in MCF-7 cells, and 12.98% and 36.21%, respectively, in MDA-MB-231 cells. 

Another series of naproxen-containing platinum(IV) prodrugs was obtained by Chen et al. [27] (Figure 9). The focus of this study was the combined action of naproxen as an inhibitor of both cyclooxygenase-2 (COX-2) and matrix metalloproteinases (MMPs) and the impact of COX and MMP inhibition on the cytotoxic activity of Pt(IV) prodrugs.

The antiproliferative activity of complexes **16**–**20** was evaluated by MTT assay on four malignant cell lines: A549, A549R (cisplatin-sensitive and cisplatin-resistant lung carcinoma, respectively), SKOV-3 (ovarian cancer), and CT-26 (colon cancer) in comparison with LO-2 (a human normal liver cell line) (Table 9). Prodrugs **19** and **20**, with two naproxen moieties in axial positions, showed lesser activity than the corresponding mono-substituted complexes **17** and **18**, respectively. The selectivity index (SI) was defined as the ratio of IC_50_ values obtained on the normal cell line LO-2 to the average of the IC_50_ values obtained on tumor cells. Cisplatin and cisplatin-based prodrug **16** showed the worst SIs of 0.5 and 0.2, respectively, while the best SI values of 1.0 and 1.5 were demonstrated by the dual-naproxen complexes **18** and **19**, respectively. The most potent compound, **17**, demonstrated higher activity than cisplatin on the cisplatin-resistant cell line A549R [27].

Overexpressed levels of matrix metallopeptidase-9 (MMP-9) are associated with cancer invasion, metastasis, and inflammation In vivo [40]. The MMP-9 inhibitory activity of prodrug **17** was evaluated by immunohistochemical staining using slices of CT-26 mice tumors in comparison with the effect of oxaliplatin and saline. Compound **17** showed significant inhibition of MMP-9 expression (6.8%), while oxaliplatin inhibition level was 8.1%, in accordance with the literature data [41]. The observed difference in inhibitory activity is associated with the presence of the naproxen moiety in prodrug **17**. However, despite high inhibitory activity towards MMP-9 expression, prodrug **17** showed low COX-inhibiting ability.

The therapeutic efficacy of prodrug **17** was assessed In vivo using a CT-26 (colon cancer) tumor model. The inhibition of tumor growth after treatment with prodrug **17** was at the level of cisplatin and oxaliplatin (317 ± 119 mm^3^, 390 ± 162 mm^3^, and 477 ± 223 mm^3^, respectively).

Recently, Jin et al. described two cisplatin-based naproxen-containing platinum(IV) prodrugs [28]. Two highly potent cisplatin prodrugs with one (NP, **10**) or two (DNP, **21**) naproxen moieties in the axial position were obtained (Figure 10). Prodrug **10** was already described by Tolan et al. [26] and synthesized following a similar procedure, while compound **21** was prepared by the reaction of oxoplatin with an excess of naproxen in the presence of 2 -(1H-benzotriazole-1-yl)-1,1,3,3-tetramethylaminium tetrafluoroborate (TBTU) and Et_3_N in N,N-dimethylformamide (DMF).

Both prodrugs **10** and **21** demonstrated outstanding antiproliferative activity towards the breast cancer cell lines MCF-7, MDA-MB-231 and MDA-MB-435 after 48 hours of incubation (Table 10). For prodrug **21**, IC_50_ values varied from 0.34 to 0.17 µM, which is up to 187-fold more than that of cisplatin, while prodrug **10** showed IC_50_ values from 1.11 to 0.4 µM. It is worth noting that the IC_50_ values obtained by Jin et al. for prodrug **10** differ significantly from the data reported by Tolan et al.; in particular, for the MCF-7 breast cancer cell line, IC_50_ values for prodrug **10** in the study by Jin et al. are 26-fold higher, and for the MDA-MB-231 triple negative breast adenocarcinoma cell line, IC_50_ values are 21-fold higher than in the paper by Tolan et al. (Table 8) [28].

The intracellular accumulation/distribution of prodrugs **10**, **21**, and cisplatin was assessed by ICP-MS assay in MCF-7 cells after incubation with 0.2 µM of each complex for 24 hours. The overall platinum content was 65- and 11-fold higher for cells incubated with **21** and **10**, respectively, than for cells incubated with cisplatin. However, less than 5% of platinum accumulated in the nucleus of cells incubated with both prodrugs, as most of the platinum was found in the cytosol and cell membrane.

The stability of prodrugs **10** and **21** in a reducing environment was assessed in a model solution in the presence of ascorbic acid or glutathione. While complex **10** showed complete degradation after 3 hours, complex **21** was still stable after 72 hours of incubation. An additional experiment in solution in the presence of 5′-GMP was conducted to verify the ability of prodrug **21** to form covalent adducts with DNA bases; no Pt-GMP or Pt-GMP_2_ were detected by ESI-MS after 6 days of incubation. This led Jin et al. to the assumption that complex **21** acts not as a prodrug, but as a whole complex that binds to DNA in a non-covalent manner [28].

Cell cycle arrest, as studied by flow cytometry, showed that prodrug **21** arrested the cell cycle of MCF-7 cells mostly in the S phase (87% at 0.2 µM incubation), while prodrug **10** arrested the cell cycle at the G2 and S phases. Further studies of prodrug **21**’s influence on the inflammatory response showed that it can greatly suppress COX-2 and programmed death-ligand 1 (PD-L1) expression in breast cancer cells, thus leading to suppressed tumor evasion from the immune system [42]. The expression of interleukins IL-1β and IL-6, which are crucial for the formation of an inflammatory response [43], was also greatly reduced by prodrug **21**.

The antitumor activity of both prodrugs was studied In vivo using mouse xenografts bearing MDA-MB-231 tumor cells. After therapy with 1.5 mg/kg of prodrug **21** for 15 days, the tumor volume was only 66.35 ± 26.07 mm^3^ (92.8% tumor growth inhibition), while after treatment with the same dose of cisplatin and saline, this value was 660 ± 68 and 926 ± 71 mm^3^, respectively. Prodrug **10** was less efficient than prodrug **21** (75.5% tumor growth inhibition), but still inhibited tumor growth better than cisplatin (227 ± 71 and 660 ± 68 mm^3^ tumor volume, respectively). No significant change in body weight was observed for the mice in all series [28].

Thus, **21** is a highly potent platinum(IV) prodrug capable of significantly suppressing tumor growth and inflammatory response.

### 2.5. Etodolac, Sulindac, and Carprofen

Three cisplatin-based platinum(IV) prodrugs with the FDA-approved NSAIDs etodolac, sulindac, and carprofen (**22–24**) were designed by Song et al. (Figure 11) [29]. Compounds were synthesized by the reaction of oxoplatin with the corresponding NSAID in the presence of TBTU and triethylamine in DMF. The cytotoxicity of prodrugs **22–24** was assessed by MTT assay. IC_50_ values of prodrug **22** were lower than the values of cisplatin on malignant cell lines MCF-7, A549 and HeLa, while being less toxic than cisplatin on the normal cell line MRC-5 (Table 11). Complex **24** showed comparable results to cisplatin toxicity on MCF-7 and A549 cell lines, while being less toxic on the HeLa cell line. The most potent prodrug, **22**, was 14-fold more active than cisplatin. Cellular accumulation in MCF-7 cells after 3 hours of incubation was assessed by ICP-MS assay, which indicated that the trend in antiproliferative activity correlated with the cellular uptake level, with prodrug **22** showing the highest level of platinum content.

Lipophilicity is considered the crucial factor for drug activity, so the logP values were obtained for cisplatin and the cisplatin-based prodrugs **22–24.** The established optimum logP values for maximum drug bioavailability is reported to be in the range of 0–3 [44,45]. Among all three prodrugs (**22**–**24**), only one logP value was in the optimum range (prodrug **22**) (Table 11). Cisplatin and prodrug **24** showed lipophilicities lower than optimum, while prodrug **23** had a logP value higher than optimum. Thus, the obtained lipophilicity values correlate with trend in cytotoxicity and cellular accumulation.

Western blot analysis conducted on MCF-7 cells incubated with prodrug **22** showed the ability of the compound to downregulate the expression of COX-2 and MDM-2 enzymes in MCF-7 cells while inducing the expression of the pro-apoptotic genes Bax and p53. Wound healing and invasion assays were conducted to assess whether prodrug **22** is able to suppress the metastasis and invasion of tumor cells. The migration rate of MCF-7 cells incubated with complex **22** was significantly lower than the rate of control and cisplatin-treated cells (18.1%, 47.7%, and 39.9%, respectively) [29].

The In vivo study of prodrug **22** on xenograft mice bearing MCF-7 tumors revealed the low systemic toxicity of prodrug **22**, and its ability to suppress tumor growth was slightly stronger than cisplatin after 14 days of treatment (457 mm^3^ and 570 mm^3^, respectively). It is notable that the suppression of tumor growth in the case of prodrug **22** was accompanied with slightly reduced platinum content in the heart, liver, and lung than in the case of cisplatin, as assessed by ICP-MS analysis of harvested organs. The body weight of mice treated with prodrug **22** remained almost unchanged, while treatment with cisplatin led to a decrease in body weight by 2 g on average, from 19 to 17 g. Thus, despite suppressing tumor growth to a higher extent than cisplatin, prodrug **22** also induced significantly fewer toxic effects [29].

## 3. Triple Action Asymmetric Pt(IV) Prodrugs with NSAIDs and Other Biologically Active Axial Ligands

The prodrug approach allows us to combine several bioactive moieties in a single molecule, which opens the way for the fine tuning of the coordination compound’s antitumor activity. The possibility of the step-by-step conjugation of axial ligands to obtain unsymmetrical Pt(IV) prodrugs opens the way for the preparation of triple action Pt(IV) prodrugs. Several papers demonstrate that upon release, all bioactive moieties exhibit their properties, leading to the increased antitumor efficiency of platinum-based drugs (Table 12) [46,47,48].

### 3.1. Indometacin/Biotin

Cisplatin-based prodrug **26**, with non-steroidal anti-inflammatory drug indomethacin, and biotin (vitamin H) with an increased affinity for tumor cells as a second axial ligand was designed by Hu et al. [49] (Figure 12). Prodrug **26** was synthesized by the reaction of oxoplatin with TBTU-activated indomethacin, followed by the reaction of the resulting complex **26a** with TBTU-activated biotin. Cisplatin-based prodrug **25**, with one indomethacin axial ligand and chlorine as the second axial ligand, was synthesized and used as a control.

The cytotoxicity of both complexes, **25** and **26**, and cisplatin as a control, were evaluated by MTT assay on HCT-116 (colorectal cancer), HepG-2 (hepatocellular carcinoma), PC-3 (prostate carcinoma), LO-2 (normal liver), EA.hy926 (umbilical vein endothelial cell), SGC7901 (gastric cancer), and SGC7901/CDDP (cisplatin-resistant gastric cancer) cell lines (Table 13). Complex **25** without the biotin moiety was the most toxic towards nearly all cell lines, while complex **26** showed lesser activity than cisplatin on nearly all malignant cells. However, prodrug **26** showed maximum toxicity towards the cisplatin-resistant cell line SGC7901/CDDP out of the three prodrugs tested, showing 9-fold higher toxicity than cisplatin. It is also worth noting that prodrug **26** was less toxic towards normal cells (LO-2 and EA.hy926; 17- and 5-fold less, respectively), than cisplatin, indicating the significant selectivity of prodrug **26** for cancer cells [49].

The intracellular accumulation/distribution of prodrugs **25** and **26** was evaluated using ICP-MS assay on PC-3, SGC7901, and SGC7901/CDDP cell lines after 12 hours of incubation (Table 14). A high level of platinum accumulation was observed for prodrug **26** in various cancer cells, surpassing the accumulation of both prodrug **25** and cisplatin. At the same time, platinum uptake level in a normal cell line (LO-2) for complex **26** was nearly two-fold lower than for cancer cells, and was similar to complex **25**. This is probably due to the higher level of biotin receptors in PC-3, SGC7901, and SGC7901/CDDP cells than in normal cells (LO-2) [49].

Both cisplatin and prodrug **25** arrested the cell cycle mainly in the G0/G1 phase, while for prodrug **26**, an increase in the cell population at the G2/M phase was observed after 72 hours of incubation in PC-3 cells. Apoptosis studies on PC-3 cells after 72 hours of incubation showed that prodrug **26** induced 30.68% and 25.01% of early and late apoptosis, respectively. Contrary to complex **26**, apoptosis induced by prodrug **25** was slightly lower (25.4% and 23.35% of early and late apoptotic cells, respectively). Also, the expression of pro- and anti-apoptotic proteins of the Bcl-2 family in PC-3 cells was studied to evaluate their role in the mechanism of cell death induced by the Pt(IV) prodrugs **25** and **26** [52]. Western blot analysis of PC-3 cells after 12 hours of incubation showed a clear increase in the expression of Bax pro-apoptotic proteins for cisplatin and prodrugs **25** and **26,** while the level of anti-apoptotic protein Bcl-2 was downregulated by both prodrugs **25** and **26** and cisplatin, with **26** showing the greatest reduction in Bcl-2 protein level.

As indomethacin is a potent non-selective COX inhibitor [53], the ability of Pt(IV) prodrugs **25** and **26** to inhibit COX-1 and COX-2 was studied in vitro via enzyme immunoassay. Both prodrugs **25** and **26** showed concentration-dependent inhibition of both COX-1 and COX-2. Interestingly, while free indomethacin showed a stronger inhibition of COX-1 than prodrugs **25** and **26**, both complexes surpassed free NSAID in terms of COX-2 inhibition. This fact is attributed to the esters of indomethacin, which have been reported to show COX-2 selectivity [54]. The authors assumed that indomethacin was not released from the prodrugs and that the Pt(IV) prodrugs **25** and **26** were able to inhibit COX-1 and COX-2 despite indomethacin not being released from the complex.

### 3.2. Ibuprofen, Aspirin/PDK, PhB, Val, and HDAC Inhibitors

In recent work published by Petruzella et al. [50], eight triple action Pt(IV) prodrugs bearing ligands with different bioactive properties were obtained. Aspirin and ibuprofen were selected as COX-2 inhibitors, dichloroacetate was chosen as a potent pyruvate dehydrogenase kinase (PDK) inhibitor, and phenylbutyrate (PhB) or valproate (Val) as histone deacetylase (HDAC) inhibitors. PDK inhibits the pyruvate dehydrogenase complex (PDHC), which plays a key role in cellular respiration [55]. In tumor cells, PDHC is inhibited, and thus cellular metabolism is shifted from glucose oxidation to glycolysis (Warburg effect) [56]. The inhibition of PDK reverses that process, leading to cell death. Inhibition of HDAC was shown to lead to chromatin de-condensation, thus making cellular DNA more sensitive to platination [57]. The synergy between HDAC inhibition and anti PD-L1 immune checkpoint blockade has been reported in several ovarian cancers [58].

Synthesis of the prodrugs **27–32** (Figure 13) was performed by two consecutive carboxylations of oxoplatin by the corresponding acid anhydride.

Stability of the prodrugs **27**–**32** in phosphate buffer was monitored by HPLC. Prodrugs **28** and **31** were stable throughout the experiment, but after the first day, prodrugs **29**, **30** and **32** showed 10%, 25% and 20% degradation, respectively. Prodrug **27** was found to be the least stable, with about half the complex degraded in the first day and the rest of the compound in the following days. The general trend was that aspirin prodrugs were less stable than ibuprofen ones [50].

The cytotoxicity of the prodrugs was studied on the various cell lines; IC_50_ values were obtained after 72 hours of incubation. All triple action prodrugs were more toxic than cisplatin, with average IC_50_ values on BCAP and PSN-1 cells being 51- and 71-fold lower, respectively, than the IC_50_ values of cisplatin All prodrugs showed nanomolar IC_50_ values on LoVo and PSN-1 cell lines and submicromolar or nanomolar IC_50_ values on other cell lines (Table 15) [50].

Two-dimensional cellular cultures are simplistic models that cannot imitate complex tumor cells processes, and thus they cannot predict the In vivo activity of a certain drug. On the other hand, 3D cell cultures possess important features of real tumors, such as intercellular interactions, hypoxia gradients, and the ability to model drug penetration and resistance [59]. These properties make cellular spheroids a relatively close model of In vivo tumors [60]. A series of triple action prodrugs (**27**–**32**) was tested against PSN-1 spheroids of pancreatic cancer. Cisplatin was used as a control. Prodrugs **31** and **32** with a valproate moiety showed sub-micromolar toxicities and were more than 50-fold more active than cisplatin. Other prodrugs in the assay were up to 15-fold more cytotoxic than cisplatin.

The absence of selectivity towards malignant cells is the core problem of platinum-based therapeutics and overcoming this issue is crucial for new anticancer drugs [61]. Thus, triple action compounds were screened on non-cancerous HEK293 cells and their selectivity index as ratio of IC_50_ (HEK293) to the IC_50_ values of cell lines that were the most sensitive to the complexes was calculated (Table 16). Prodrug **32**, with valproate and aspirin moieties, showed high selectivity against BCPAP cells (SI = 59); prodrug **31**, with valproate and ibuprofen moieties, was moderately selective on LoVo cells (SI = 13.8); while the other prodrugs did not show high selectivity towards any of the tumor cells (ranging from 0.1 to 2).

Platinum drugs exhibit their cytotoxic action through DNA platination; thus, platinum cellular accumulation and DNA platination are key parameters that define the efficiency of platinum drugs. To investigate whether the increased cytotoxicity of prodrugs **27–32** is due to increased platinum uptake and better cisplatin binding to DNA, cellular uptake and DNA platination were evaluated on PSN-1 cells after 24 hours of incubation with 1 µM of each prodrug (**27–32**) and cisplatin (Table 17). The results indicate that there is no correlation between cellular uptake and cytotoxic activity. The most potent prodrug, **27**, showed the lowest level of platinum in PSN-1 cells, while prodrugs with similar IC_50_ values (**28** and **31**) showed the highest values of platinum uptake. No correlation between toxicity and nuclear DNA platination was observed either. Prodrug **30** showed the highest platination level; however, it is the least active prodrug in the series [50].

The ability of prodrugs **27**–**32** to inhibit HDAC was evaluated in PSN-1 cells (Table 16). Prodrugs **29–32** with PhB and Val in the axial positions showed high inhibitory activities towards HDAC; however, no correlation between the inhibition of HDAC and toxicity was observed.

The COX-2 inhibition ability of prodrugs **27–32** on the PSN-1 cell line was studied as well. Surprisingly, ibuprofen-containing prodrugs **28**, **30**, and **31** showed high inhibitory activity, contrary to the results in previous reports (Section 2.1) with the ibuprofen–platinum(IV) prodrugs **3** and **5**. Furthermore, prodrug **31** turned out to be the most potent COX inhibitor in the series (11% inhibition), while aspirin-containing prodrug **32** was the least active [50].

Dichloroacetate is reported to cause imbalances in redox homeostasis, leading to reactive oxygen species (ROS) production and damage to mitochondria [62]. Thus, the activity of the dichloroacetate moiety in prodrugs **27** and **28** was assessed by measuring intracellular ROS level and changes in mitochondrial membrane potential. Val-containing prodrugs **31** and **32** caused significant increases in intracellular ROS level in PSN-1 cells despite the absence of a dichloroacetate (DCA) ligand, while PSN-1 cells incubated with the DCA-containing prodrugs **27** and **28** demonstrated a lower level of ROS in PSN-1 cells (Table 16). An increase in intracellular ROS level leads to hypopolarization of the mitochondrial membrane [63]. Thus, a change in the membrane mitochondrial potential was evaluated in PSN-1 cells incubated with prodrugs **27–32.** All triple action prodrugs showed a higher mitochondrial membrane depolarization ability than cisplatin, with prodrugs **29** and **31** causing the highest hypopolarization (25–40%) (Table 16). Interestingly, complexes **29–32** with no mitochondria-active moieties showed significant mitochondrial depolarization activity as well.

Thus, no simple correlation between a certain toxicity mechanism and the biological activity of triple action prodrugs **27–32** was revealed. More likely, a complex synergistic effect based on specific cellular interactions of each of the components is taking place [50].

### 3.3. Aspirin/Estramustine

A large class of potent anticancer drugs exhibit their antiproliferative action by disrupting mitotic spindle assembly [64]. Paclitaxel and docetaxel act as microtubule-targeting agents (MTAs), inhibiting tubulin activity which leads to the disruption of intracellular transport and inhibition of cell proliferation [65]. A number platinum(IV) prodrugs with MTAs have been already described [66,67]. Recently, Karmakar et al. [51] have reported on a series of platinum(IV) prodrugs with estramustine, an FDA-approved MTA for the palliative treatment of metastatic prostate carcinoma. The designed series consist of unsymmetrical prodrugs with estramustine and various other inhibitors, such as HDAC inhibitors, PDK inhibitors, and COX inhibitors, in axial positions. For the scope of this review, only aspirin-containing Pt(IV) prodrug **33** is considered (Figure 14).

Prodrug **33** was synthesized via several steps. First, estramustine was reacted with succinic anhydride to obtain estramustine succinate, which was then activated by N-hydroxysuccinimide (NHS). The estramustine-cisplatin conjugate **33a** was obtained by reacting oxoplatin with the NHS-ester of estramustine succinate. Prodrug **33** was synthesized by the carboxylation of the **33a** by aspirin anhydride. Prodrug **33** was found to be stable in phosphate buffer with a half-life of several days. The analysis of prodrug **33** reduction in phosphate buffer by 10 equivalents of ascorbic acid determined that half of the complex is reduced in 4.2 hours. [51].

The cytotoxicity of **33** was assessed by MTT assay, with cisplatin, free estramustine, and prodrug **33a** as controls (Table 18). While cisplatin and the free ligand showed IC_50_ values in the range of 6–26 µM, for prodrug **33a**, IC_50_ values were in the sub-micromolar range (0.19–0.9 µM). The triple action drug **33**, however, was found to be much more toxic on all cell lines compared to cisplatin and prodrug **33a,** with toxicities ranging from 0.09 to 0.45 µM. The IC_50_ values were also determined for the non-cancerous cell line MCR-5 pd30 to assess the selectivity of the prodrug. The selectivity index was determined against LNCaP and DU145 prostate cancer cells. Prodrug **33** showed low toxicity (2.2 µM) and high selectivity towards malignant cells. The selectivity index of prodrug **33** was more than 18, while for control compounds (cisplatin and free estramustine), it did not surpass 3 (although for **33a**, the SI was close to that of **33** (17.7 and 18.3, respectively)) [51].

The intracellular accumulation of prodrug **33** in LNCaP cells was evaluated by ICP-MS assay. Compared to cisplatin, intracellular platinum level for prodrug **33** was 46-fold higher after 6 hours of incubation with 10 µM of the compound (Table 19).

LogP values of cisplatin and **33** were determined by the shake-flask method. When the lipophilicities of other prodrugs in the series had been considered, a clear correlation between lipophilicity and platinum uptake was observed. Based on these results, the authors assumed that the prodrugs were able to enter the cells intact without being reduced in the extracellular medium [51].

The study of the cell cycle arrest caused by prodrug **33** in LNCaP cells was utilized to determine whether both estramustine and cisplatin contribute to the mechanism of action of prodrug **33**. Estramustine was found to stop the cell cycle mainly in the G2/M phase due to its antitubulin activity, while cisplatin mainly arrested the cell cycle in the S phase (and partly in the G1 phase). For prodrug **33**, the cell cycle was arrested mainly in the G1 phase, with a significant share of the S and G2/M phases. Thus, the effect of both bioactive moieties on cell cycle arrest was clearly shown [51].

Autophagy is a process used by cells to degrade damaged or unneeded organelles. NSAIDs, notably aspirin, are capable of inducing autophagy in HCT116 human colon cancer cells [68]. Thus, the ability of prodrug **33** to induce autophagy was investigated by Western blot analysis of the protein extracts from LNCaP cells. Prodrug **33** showed an ability to induce autophagy in LNCaP cells in a concentration-dependent manner, which was seen from the expression levels of p62, which was efficiently degraded by autophagy. On the other hand, the estramustine–cisplatin prodrug **34**, with acetyl as the second axial ligand, did not induce any changes in p62 expression.

Thus, a highly potent triple action platinum(IV) prodrug **33** with estramustine and aspirin as axial ligands was obtained. The prodrug **33** was found to be up to 110-fold more cytotoxic than cisplatin and up to 80-fold more cytotoxic than estramustine, suggesting a synergetic effect. At the same time, the prodrug showed high selectivity toward prostate cancer, as it was 18-fold less toxic on MRC-5 pd30 normal cells than on LNCaP and DU145 on average. This selectivity was not observed for either estramustine or cisplatin. A study on the mechanism of action of prodrug **33** determined that it acts as a multiple-action prodrug, with all three ligands affecting multiple targets in cancer cells: both cisplatin and estramustine moieties act as antiproliferative agents, while aspirin acts as an autophagy-inducing agent [51].

## 4. Conclusions

Platinum(IV) prodrugs are one of the most promising classes of novel platinum anticancer agents. Octahedral Pt(IV) complexes are considered to be kinetically inert, while the axial position can be easily modified. Thus, Pt(IV) prodrugs can be finely tuned, which facilitates the development of antitumor agents with the desired mode of action. In this review, platinum(IV) prodrugs with NSAIDs in the axial position are considered.

The design of platinum(IV) prodrugs with FDA-approved agents in the axial position is a promising approach due to the well-known pharmacological profile and biocompatibility of anti-inflammatory agents. The conjugation of hydrophilic cisplatin with lipophilic NSAIDs results in a high-lipophilic Pt(IV) prodrug capable of effective cellular penetration and ideally, of slow release of the NSAID and the cytotoxic agent intracellularly. A significant increase in intracellular penetration of Pt(IV) prodrugs compared to the initial Pt(II) drugs expectedly leads to impressive increases in antiproliferative activity; most of the prodrugs considered in this review demonstrated IC_50_ values more than 10-fold lower than cisplatin, with prodrugs **3**, **21**, and **27–32** being from 180- to 300-fold more potent than cisplatin. LogP values obtained for several prodrugs demonstrate a clear correlation between intracellular accumulation, lipophilicity, and antiproliferative activity; thus, lipophilicity seems to be a key factor that determines the efficiency of Pt(IV) prodrugs. The trend between lipophilicity and antiproliferative activity, as demonstrated by Ravera et al. for prodrugs **8** and **9**, as well as by Song et al. for prodrugs **22–24** and by Karmakar et al. for prodrug **33**, support this assumption.

COX inhibition activity has little to no influence on cytotoxicity; the extremely potent prodrug **3** shows no inhibition activity on either COX isoform, and no correlation between COX expression and cytotoxicity is observed for prodrugs **8** and **9**. However, several Pt(IV) prodrugs considered in this review, such as **1**, **2**, **21** and **31**, act as potent anti-inflammatory agents and can probably reduce the inflammation-induced side effects of platinum chemotherapy; i.e., act as true double-acting drugs.

The step-by-step carboxylation of oxoplatin with different bioactive moieties opens the way for the design of multi-action platinum (IV) prodrugs. Several triple action Pt(IV) prodrugs with NSAIDs and other bioactive moieties in the axial position are considered in this review. Biochemical assays for prodrugs **26–33** confirmed that each prodrug acts as a triple action compound, possessing biological properties of each of its moieties. In addition, the resulting antiproliferative activity of such prodrugs was up to 300-fold more than the antiproliferative activity of cisplatin. Contrary to dual action prodrugs with NSAIDs **1–24**, no simple correlation between cytotoxicity and one single parameter was observed for triple action prodrugs **27–32**. More likely, a complex synergetic effect of all moieties acting simultaneously is taking place. The design of triple-action Pt(IV) prodrugs is a new, promising approach which opens up new possibilities for the use of Pt(IV) prodrugs as chemotherapeutic agents; the lack of data on the therapeutic efficacy of such derivatives In vivo is disappointing.

The results of the In vivo antitumor efficiency of Pt(IV) prodrugs with NSAIDs as axial ligands published to date (namely **17**, **21** and **22**) do not allow us to derive exact conclusions. Prodrugs **17** and **22** showed tumor growth inhibition values close to cisplatin on cisplatin-sensitive CT-26 and MCF-7 tumor models, respectively, while prodrug **21** was 10-fold more active In vivo than cisplatin in a cisplatin-resistant MDA-MB-231 tumor model. However, a common trend was observed for all three prodrugs (**17, 21** and **22**) tested In vivo; the body weight of mice treated with the prodrugs decreased little to none, while the conventional platinum(II) drugs cisplatin and oxaliplatin caused significant body weight loss. Thus, although the general therapeutic efficiency of NSAID-containing platinum(IV) prodrugs is still unclear, it is undoubtable that platinum(IV) prodrugs reduce the side effects and general toxicity of platinum chemotherapy.

## Figures and Tables

**Figure 1 ijms-22-03817-f001:**
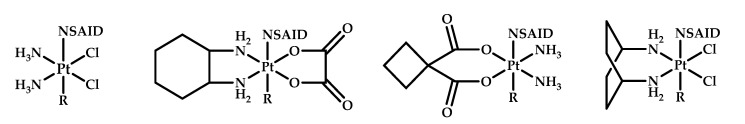
General structures of cisplatin-based, oxaliplatin-based, carboplatin-based, and kiteplatin-based Pt(IV) prodrugs with non-steroidal anti-inflammatory drugs (NSAIDs) as axial ligand(s).

**Figure 2 ijms-22-03817-f002:**
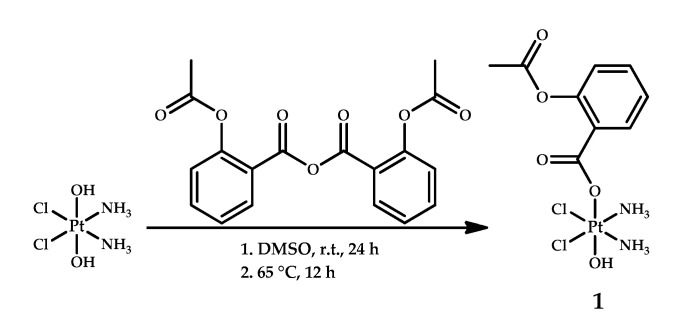
Synthesis of Platin A (asplatin) **1**, the first cisplatin-based Pt(IV) prodrug with a NSAID as an axial ligand. DMSO -dimethylsulfoxide; r.t.—room temperature.

**Figure 3 ijms-22-03817-f003:**
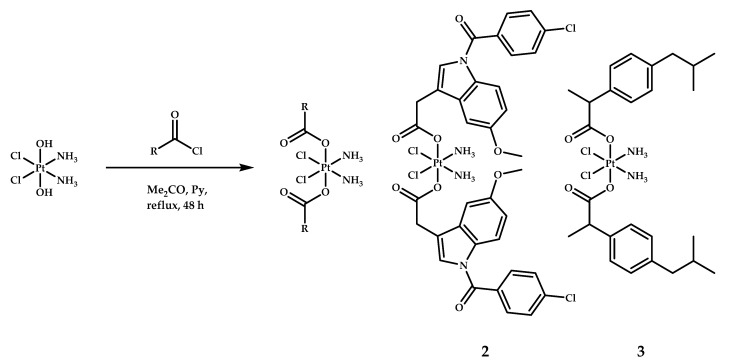
Synthesis of platinum(IV) prodrugs **2** and **3** with indomethacin and ibuprofen as the axial ligands.

**Figure 4 ijms-22-03817-f004:**
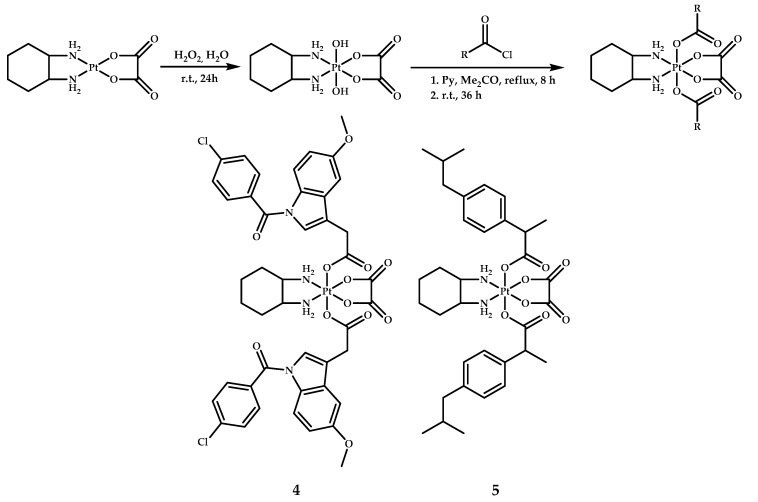
Synthesis of oxaliplatin-based platinum(IV) prodrugs **4** and **5** with indomethacin and ibuprofen as axial ligands.

**Figure 5 ijms-22-03817-f005:**
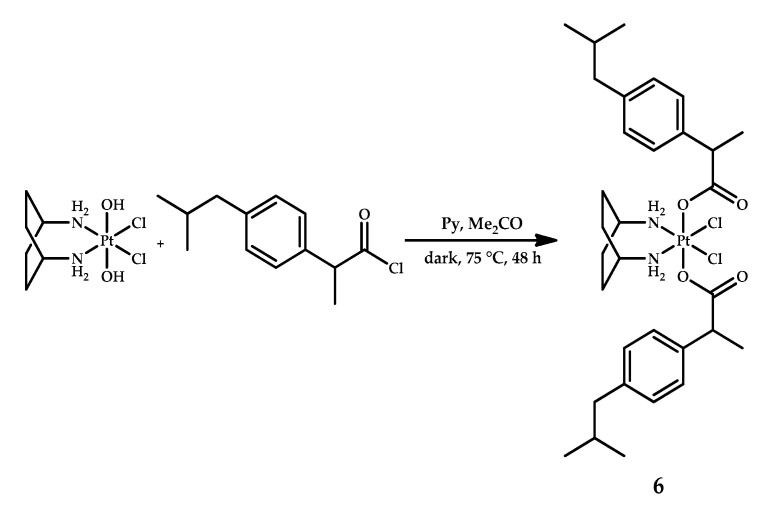
Synthesis of kiteplatin-based platinum(IV) prodrug **6** with two ibuprofen moieties as axial ligands.

**Figure 6 ijms-22-03817-f006:**
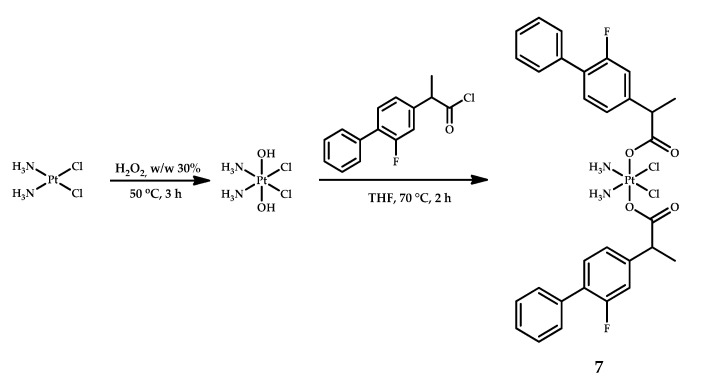
Synthesis of cisplatin-based prodrug **7** with two flurbiprofen moieties as axial ligands.

**Figure 7 ijms-22-03817-f007:**
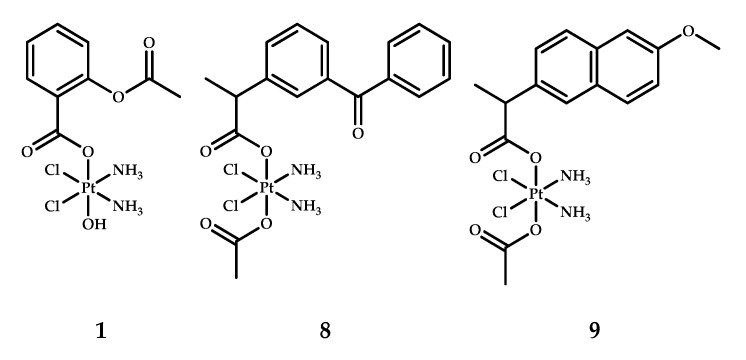
Asplatin **1** and cisplatin-based platinum(IV) prodrugs **8** and **9** with ketoprofen and naproxen as axial ligands.

**Figure 8 ijms-22-03817-f008:**
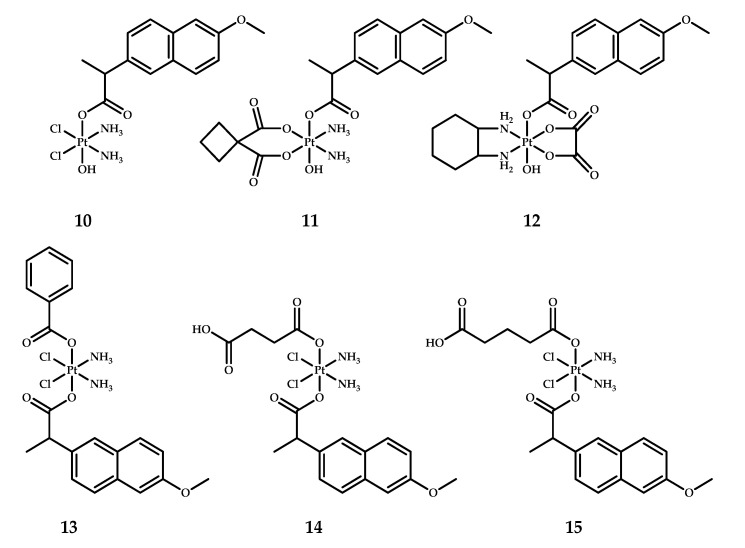
Naproxen-containing mono-carboxylated platinum(IV) prodrugs based on cisplatin **10**, carboplatin **11** and oxaliplatin **12** and di-carboxylated **13–15**.

**Figure 9 ijms-22-03817-f009:**
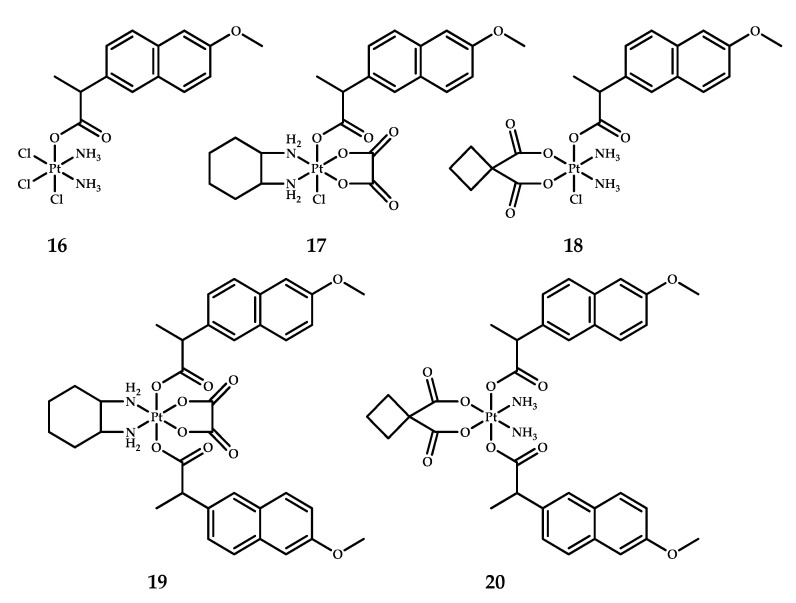
Naproxen-containing platinum(IV) prodrugs based on cisplatin **16**, carboplatin **17** and oxaliplatin **18** and di-carboxylated complexes based on oxaliplatin **19** and carboplatin **20**.

**Figure 10 ijms-22-03817-f010:**
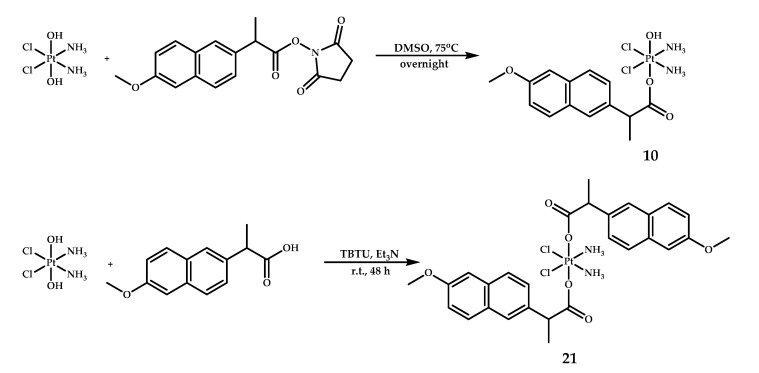
Synthesis of cisplatin-based monocarboxylated platinum(IV) prodrug **10** and dicarboxylated platinum(IV) prodrug **21** with one or two moieties of naproxen in axial positions, respectively.

**Figure 11 ijms-22-03817-f011:**
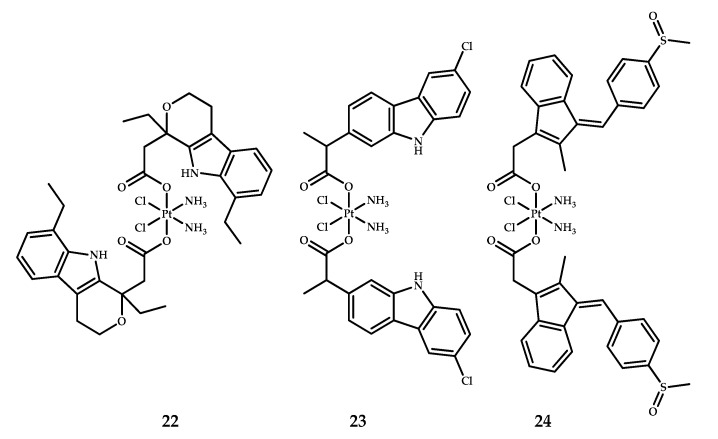
Platinum(IV) prodrugs with etodolac **22**, carprofen **23** and sulindac **24** moieties as axial ligands.

**Figure 12 ijms-22-03817-f012:**
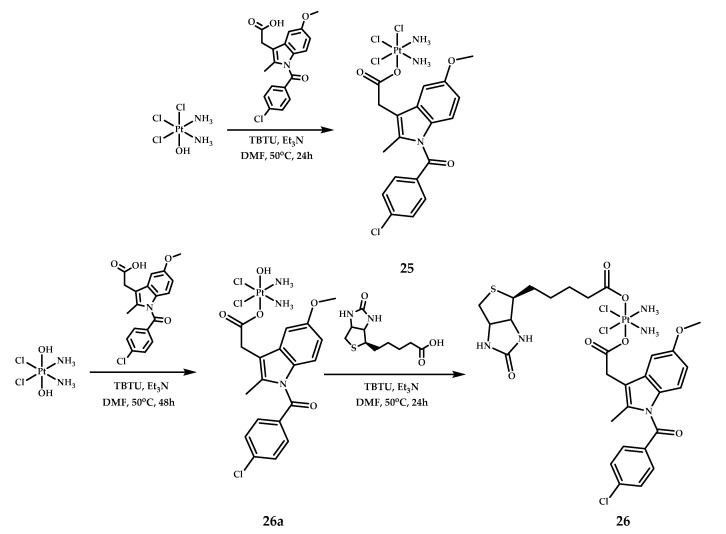
Cisplatin-based platinum(IV) prodrugs **25**, **26a**, and **26** with indomethacin and biotin as axial ligands.

**Figure 13 ijms-22-03817-f013:**
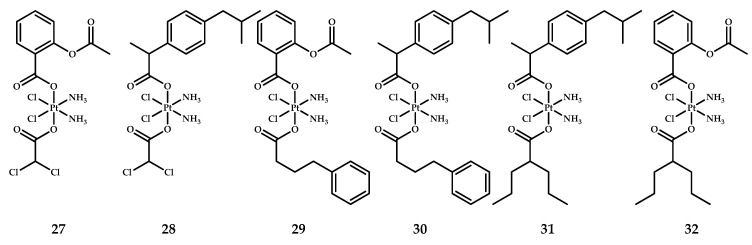
Cisplatin-based triple action prodrugs **27–32** with ibuprofen (**28**, **30**, **31**), aspirin (**27**, **29**, **32**), dichloroacetate (**27, 28**), phenylbutyrate (**29, 30**) and valproate (**31**, **32**) moieties in axial positions.

**Figure 14 ijms-22-03817-f014:**
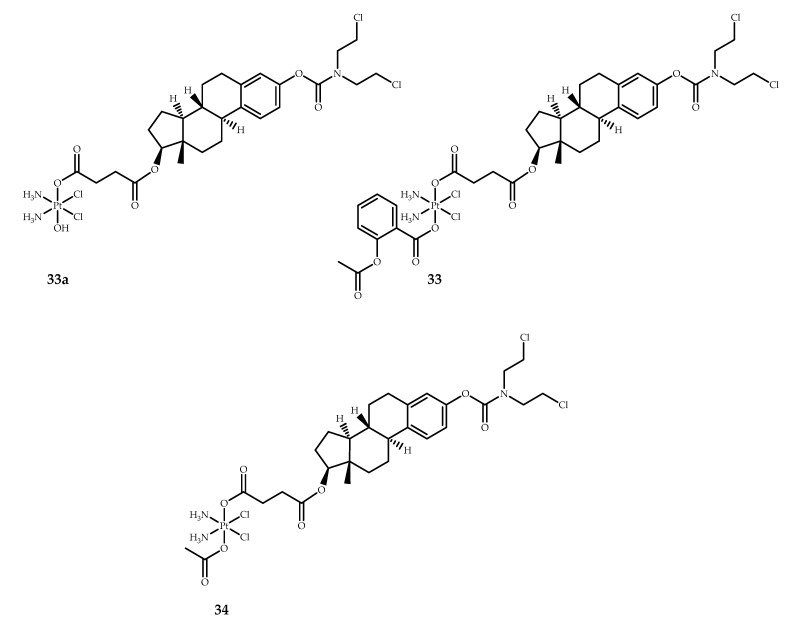
Cisplatin-based platinum(IV) prodrugs **33a, 33**, and **34** with an estramustine moiety in the axial position. Prodrug **33** contains an aspirin moiety in the axial position.

**Table 1 ijms-22-03817-t001:** Dual action platinum(IV) prodrugs with NSAIDs as axial ligand(s). DU145—androgen independent human prostate adenocarcinoma cells; HNSCC—Head and neck squamous cell carcinoma; MDA-MB-231—triple negative human breast adenocarcinoma; HCT-116—human colon carcinoma; BEL7404—human hepatoma cell line; A2780—human ovarian carcinoma; MCF-7—human breast adenocarcinoma; CT-26—murine colorectal carcinoma.

Compound	Pt(II) Drug	NSAID	R (Second Axial Position)	IC_50_ (Tumor Cell Type)	In Vivo Studies (Tumor, Dose, Tumor Growth Suppression, Day of Treatment)	Reference
**1**	Cisplatin	Aspirin	Hydroxyl	8 ± 3 (DU145)	-	[20]
**2**	Cisplatin	Indometacin	Indometacin	0.69 (1483 HNSCC)	-	[21]
**3**	Cisplatin	Ibuprofen	Ibuprofen	0.045 (1483 HNSCC)		
**4**	Oxaliplatin	Indometacin	Indometacin	0.55 (MDA-MB-231)	-	[22]
**5**	Oxaliplatin	Ibuprofen	Ibuprofen	0.33 (MDA-MB-231)		
**6**	Kiteplatin	Ibuprofen	Ibuprofen	0.26 ± 0.03 (HCT116)	-	[23]
**7**	Cisplatin	Flurbiprofen	Flurbiprofen	1.4 ± 1.1 (BEL7404)	-	[24]
**8**	Cisplatin	Ketoprofen	Acetyl	0.063 ± 0.033 (A2780)	-	[25]
**9**	Cisplatin	Naproxen	Acetyl	0.045 ± 0.033 (A2780)		
**10**	Cisplatin	Naproxen	Hydroxyl	10.40 ± 0.79 (MCF-7)	-	[26]
**11**	Carboplatin	Naproxen	Hydroxyl	9.12 ± 0.63 (MCF-7)		
**12**	Oxaliplatin	Naproxen	Hydroxyl	9.47 ± 0.75 (MCF-7)		
**13**	Cisplatin	Naproxen	Benzoic acid	3.92 ± 0.42 (MCF-7)		
**14**	Cisplatin	Naproxen	Succinic acid	7.65 ± 0.84 (MCF-7)		
**15**	Cisplatin	Naproxen	Glutaric acid	8.73 ± 0.89 (MCF-7)		
**16**	Cisplatin	Naproxen	Cl	0.2 ± 0.1 (CT-26)	**17:** CT-26 4 mg/kg 82.5% 15 days	[27]
**17**	Oxaliplatin	Naproxen	Cl	2.9 ± 0.7 (CT-26)		
**18**	Carboplatin	Naproxen	Cl	26.1 ± 8.6 (CT-26)		
**19**	Oxaliplatin	Naproxen	Naproxen	8.2 ± 0.6 (CT-26)		
**20**	Carboplatin	Naproxen	Naproxen	27.3 ± 5.7 (A549)		
**21**	Cisplatin	Naproxen	Naproxen	0.16 ± 0.01 (MDA-MB-231)	**21:** MDA-MB-231; 1.5 mg/kg 92.8% 15 days	[28]
**10**	Cisplatin	Naproxen	Hydroxyl	0.40 ± 0.10 (MCF-7)		
**22**	Cisplatin	Etodolac	Etodolac	0.17 ± 0.04 (MCF-7)	**22:** MCF-7; 3 mg/kg 15 days, 60.6%	[29]
**23**	Cisplatin	Carprofen	Carprofen	0.95 ± 0.04 (MCF-7)		
**24**	Cisplatin	Sulindac	Sulindac	2.68 ± 1.09 (MCF-7)		

**Table 2 ijms-22-03817-t002:** Antiproliferative activity of cisplatin, asplatin **1,** and an equimolar mixture of cisplatin + aspirin on the prostate tumor cell lines PC3 (androgen independent human prostate adenocarcinoma cells), DU145(androgen independent human prostate adenocarcinoma cells), and LNCaP (androgen-sensitive human prostate adenocarcinoma cells).

Compound/Cell Line	IC_50_, µM		
	PC3	DU145	LNCaP
**Cisplatin**	14 ± 4	5 ± 2	15 ± 1
**Asplatin 1**	15 ± 5	8 ± 3	12 ± 1
**Aspirin + Cisplatin (1:1)**	14 ± 6	4 ± 1	18 ± 1

**Table 3 ijms-22-03817-t003:** Antiproliferative activity of cisplatin and platinum(IV) prodrugs **2** and **3** on HCT-116 (colorectal carcinoma), OVCAR3 (ovarian cancer), MDA-MB-231 (triple negative breast cancer), and 1483 HNSCC (head and neck squamous cell carcinoma) cell lines. The ability of cells to express cyclooxygenase-2 (COX-2) is indicated by “−” (non-expressing), “+” (expressing), “++” (strongly expressing).

Compound/Cell Line	IC_50_, µM			
	HCT-116	OVCAR3	MDA-MB-231	1483 HNSCC
COX-2 expression	−	+	+	++
**Cisplatin**	12	2.07	20	2
**2**	1.1	2.2	1.65	0.69
**3**	0.065	0.13	0.05	0.045

**Table 4 ijms-22-03817-t004:** Antiproliferative activity of oxaliplatin-based platinum(IV) prodrugs **4** and **5** and cisplatin on HCT-116 (colorectal carcinoma) and MDA-MB-231 (triple negative breast adenocarcinoma) cell lines. The ability of cells to express cyclooxygenase-2 (COX-2) is indicated by “−” (non-expressing), “+” (expressing).

Compound/Cell Line	IC_50_, µM	
	HCT-116	MDA-MB-231
COX-2 expression	-	+
**Cisplatin**	12	20
**4**	1.3	0.55
**5**	0.31	0.33

**Table 5 ijms-22-03817-t005:** Cellular accumulation of prodrugs **2–5** and cisplatin (µg Pt/10^6^ cells) on HCT-116 (colorectal carcinoma) and MDA-MB-231 (triple negative adenocarcinoma) cell lines.

	Cellular Uptake, µg Pt/10^6^ Cells	
Compound/Cell line	HCT-116	MDA-MB-231
**Cisplatin**	0.008 ± 0.001	0.013 ± 0.009
**2**	0.26 ± 0.04	0.4 ± 0.1
**3**	0.096 ± 0.007	0.17 ± 0.03
**4**	0.014 ± 0.001	0.033 ± 0.009
**5**	0.18 ± 0.01	0.27 ± 0.03

**Table 6 ijms-22-03817-t006:** Antiproliferative activity of kiteplatin, cisplatin, ibuprofen, and platinum prodrug **6** on HCT15 (human colon carcinoma) and HCT116 (colon adenocarcinoma) cells.

	IC_50,_ µM	
Compound/Cell Line	HCT15	HCT116
**6**	0.45 ± 0.04	0.26 ± 0.03
**Kiteplatin**	11 ± 1	7 ± 1
**Cisplatin**	17 ± 2	11 ± 1
**Ibuprofen**	>800	708 ± 8

**Table 7 ijms-22-03817-t007:** Antiproliferative activity of Pt(IV) prodrug **7** on SW480 (colon adenocarcinoma), PC-3 (prostate adenocarcinoma), PANC-1 (pancreatic cancer), A549, A549-DDP (lung carcinoma), BEL7404 (liver carcinoma), and BEL7404-CP20 (cisplatin-resistant liver carcinoma) cell lines.

Compound/Cell Line	IC_50_, µM						
	SW480	PC-3	PANC-1	A549	A549-DDP	BEL7404	BEL7404-CP20
**Cisplatin**	49 ± 1.1	21.2 ± 1.1	14.4 ± 1.1	7.4 ± 1.0	20.03 ± 1.1	14.7 ± 1.1	>50
**Cisplatin–Flurbiprofen (1:2)**	29.6 ± 1.0	22.4 ± 1.1	11.1 ± 1.1	7.1 ± 1.0	21.1 ± 1.2	18.1 ± 1.0	>50
**7**	0.6 ± 1.1	3.4 ± 1.0	3.4 ± 1.1	2.7 ± 1.1	2.5 ± 1.1	1.4 ± 1.1	3.1 ± 1.1

**Table 8 ijms-22-03817-t008:** Antiproliferative activity of cisplatin-based prodrugs **1**, **8** and **9** on A549 (lung carcinoma), HT-29 (colon cancer), HCT 116 (colon adenocarcinoma), SW480 (colorectal cancer), A2780 (ovarian cancer), and MSTO-21 (malignant pleural mesothelioma) cell lines.

Compound/Cell Line		IC_50_, µM					
		A549	HT-29	HCT 116	MSTO-211H	SW480	A2780
	Log k’	COX Expression					
		++	++	+	-	-	-
**Cisplatin**	−0.5	3.60 ± 0.90	2.72 ± 0.39	3.05 ± 0.28	1.33 ± 0.35	2.27 ± 0.12	0.46 ± 0.11
**Oxaliplatin**	−0.28	0.74 ± 0.25	0.92 ± 0.08	1.16 ± 0.09	1.01 ± 0.55	0.48 ± 0.02	0.171 ± 0.008
**1**	−0.32	6.40 ± 2.7	4.42 ± 0.21	1.50 ± 0.083	1.74 ± 0.21	0.217 ± 0.07	0.552 ± 0.123
**8**	0.14	0.825 ± 0.388	0.486 ± 0.235	0.184 ± 0.088	0.198 ± 0.035	0.0948 ± 0.023	0.063 ± 0.033
**9**	0.18	0.486 ± 0.075	0.313 ± 0.186	0.149 ± 0.076	0.161 ± 0.040	0.0844 ± 0.0287	0.045 ± 0.016

**Table 9 ijms-22-03817-t009:** Antiproliferative activity of cisplatin-, carboplatin-, oxaliplatin-based and platinum(IV) prodrugs **16–20** on A549, A549R (cisplatin-sensitive and cisplatin-resistant lung carcinoma, respectively), SKOV-3 (ovarian cancer), CT-26 (colon cancer), and LO-2 (human normal liver) cell lines. Selectivity index (SI) = IC_50_(LO-2)/average IC_50_ values.

Compound/Cell Line	IC_50_, µM					
	A549	A549R	SKOV-3	CT-26	LO-2	SI^a^
**Cisplatin**	4.8 ± 0.6	15.1 ± 1.1	2.5 ± 0.4	0.3 ± 0.1	3.0 ± 0.7	0.5
**Oxaliplatin**	8.4 ± 2.2	7.3 ± 1.7	9.4 ± 2.3	2.30 ± 0.3	3.6 ± 0.5	0.5
**Carboplatin**	79.6 ± 18.4	60.6 ± 14.7	38.1 ± 9.6	46.2 ± 11.4	70.7 ± 16.3	-
**16**	2.2 ± 0.3	19.7 ± 2.5	14.4 ± 0.7	0.2 ± 0.1	1.9 ± 0.4	0.2
**17**	5.2 ± 0.5	4.8 ± 0.3	8.5 ± 0.1.6	2.9 ± 0.7	4.8 ± 0.8	0.9
**18**	47.2 ± 6.9	62.2 ± 16.7	26.0 ± 5.5	26.1 ± 8.6	39.9 ± 9.8	1.5
**19**	10.2 ± 1.0	12.0 ± 0.4	11.3 ± 0.8	8.2 ± 0.6	15.3 ± 2.5	0.5
**20**	27.3 ± 5.7	83.0 ± 16.4	48.9 ± 7.5	48.9 ± 8.4	26.3 ± 4.7	0.8

**Table 10 ijms-22-03817-t010:** Antiproliferative activity of cisplatin, equimolar mixtures of cisplatin and naproxen, and platinum(IV) prodrugs **21** and **10** on MCF-7 (breast cancer), MDA-MB-231 (triple negative breast adenocarcinoma), and MDA-MB-435 (triple negative breast adenocarcinoma) cell lines.

Compound/Cell Line	IC_50_, µM		
	MCF-7	MDA-MB-231	MDA-MB-435
**21**	0.17 ± 0.04	0.16 ± 0.01	0.34 ± 0.09
**10**	0.40 ± 0.10	0.81 ± 0.02	1.11 ± 0.06
**Cisplatin**	4.00 ± 1.00	29.98 ± 1.10	8.34 ± 0.49
**Cisplatin + Naproxen**	7.00 ± 2.00	>64	>32
**Cisplatin + 2 Naproxen**	5.18 ± 2.80	>64	>32
**Naproxen**	>64	>64	>64

**Table 11 ijms-22-03817-t011:** Antiproliferative activity of cisplatin and prodrugs **22–24** on MCF-7 (breast cancer), A549 (lung carcinoma), HeLa (cervical cancer), and MCR-5 (embryo fibroblast) cell lines. LogP values and cellular uptake in MCF-7 cells are also shown.

Compound/Cell Line	IC_50_, µM				LogP	Cellular Uptake, ng Pt/10^6^ Cells
	MCF-7	A549	HeLa	MRC-5		
**Cisplatin**	13.31 ± 2.90	11.29 ± 0.21	9.82 ± 0.52	4.92 ± 0.13	−4.86	200
**22**	0.95 ± 0.42	2.78 ± 0.28	3.59 ± 0.17	6.24 ± 1.10	2.56	7000
**23**	2.68 ± 1.09	3.87 ± 1.03	2.99 ± 0.07	7.06 ± 0.91	3.92	2100
**24**	9.02 ± 2.83	10.64 ± 0.06	42.33 ± 7.52	14.04 ± 3.04	−0.42	1900

**Table 12 ijms-22-03817-t012:** Triple action platinum(IV) prodrugs with NSAIDs and another bioactive ligands in the axial position.

Compound	COX-Inhibitor	Second Axial Ligand	Confirmation of Second Bioactive Ligand Action	Reference
**26**	Indomethacin	Biotin (biotin receptors targeting)	Selectivity to tumor cells is attributed to increased amount of biotin receptors SMVT	[49]
**27**	Aspirin	Dichloroacetate (DCA) (PDK inhibitor)	MMP depolarization	[50]
**28**	Ibuprofen		MMP depolarization	
**29**	Aspirin	Phenylbutyrate (PhB) (HDAC inhibitor)	HDAC inhibition	
**30**	Ibuprofen		HDAC inhibition	
**31**	Ibuprofen	Valproate (Val) (HDAC inhibitor)	HDAC inhibition	
**32**	Aspirin		HDAC inhibition	
**33**	Aspirin	Estramustine (tubulin inhibitor)	Increase in cell population arrested in G2/M phase compared to cisplatin. Free estramustine is known to arrest the cell cycle in G2/M	[51]

**Table 13 ijms-22-03817-t013:** Antiproliferative activity of cisplatin and platinum(IV) prodrugs **25** and **26** on HCT-116 (colorectal cancer), HepG-2 (hepatocellular carcinoma), PC-3 (prostate carcinoma), LO-2 (normal liver), EA.hy926 (umbilical vein endothelial cell), SGC7901 (gastric cancer), and SGC7901/CDDP (cisplatin-resistant gastric cancer) cell lines. Rf: ratio of IC_50_ value for a cisplatin-resistant cell line and IC_50_ value for a cisplatin-sensitive line (IC_50_ SGC7901/CDDP / IC_50_ SGC7901).

Compound/Cell Line	IC_50_, µM							
	HCT-116	HepG-2	PC-3	LO-2	EA.hy926	SGC7901	SGC7901/CDDP	Rf
**Cisplatin**	7.78 ± 0.63	3.96 ± 0.28	0.95 ± 0.07	3.54 ± 0.26	7.42 ± 0.36	1.11 ± 0.09	8.18 ± 0.73	7.37
**25**	4.94 ± 0.37	2.35 ± 0.18	0.81 ± 0.07	4.79 ± 0.32	9.61 ± 0.23	1.36 ± 0.09	4.50 ± 0.41	3.31
**26**	19.27 ± 1.4	9.67 ± 0.84	7.24 ± 0.26	59.64 ± 2.32	41.73 ± 2.1	3.27 ± 0.26	0.91 ± 0.06	0.29

**Table 14 ijms-22-03817-t014:** Cellular uptake in PC-3 (prostate carcinoma), SGC7901 (gastric cancer), SGC7901/CDDP (cisplatin-resistant gastric cancer), and LO-2 (normal liver) cell lines preincubated with prodrugs **25**, **26**, and cisplatin after 12 hours of incubation (ng/10^6^ cells).

Compound/Cell Line	Cellular Uptake, ng/10^6^ Cells			
	PC-3	SGC7901	SGC7901/CDDP	LO-2
**Cisplatin**	133	120	28	141
**25**	276	451	269	372
**26**	839	790	757	403

**Table 15 ijms-22-03817-t015:** Antiproliferative activity of cisplatin and platinum(IV) prodrugs **27–32** on HCT-15 (human colon carcinoma), BCAP (thyroid cancer), PSN-1 (pancreatic cancer), LoVo (colon cancer), 2008 (ovarian adenocarcinoma), and C13* (cisplatin-resistant ovarian adenocarcinoma) cell lines. Resistance factor (RF) = IC_50_ (C13*)/IC_50_ (2008).

Class	IC_50_, µM							
	Compound/Cell Line	HCT-15	BCPAP	PSN-1	LoVo	2008	C13*	RF
PDKi, COXi	**27**	1.03 ± 0.25	0.06 ± 0.005	0.06 ± 0.008	0.755 ± 0.06	0.61 ± 0.19	1.66 ± 0.09	2.7
	**28**	0.65 ± 0.17	0.14 ± 0.04	0.08 ± 0.01	0.285 ± 0.02	0.32 ± 0.09	0.97 ± 0.22	3
HDACi, COXi	**29**	1.86 ± 0.41	0.06 ± 0.004	0.09 ± 0.02	0.055 ± 0.01	0.29 ± 0.11	0.43 ± 0.12	1.5
	**30**	4.98 ± 1.25	0.08 ± 0.01	0.92 ± 0.2	0.211 ± 0.08	0.89 ± 0.19	1.65 ± 0.11	1.9
	**31**	3.98 ± 0.89	0.68 ± 0.08	0.07 ± 0.01	0.034 ± 0.03	1.35 ± 0.29	1.61 ± 0.42	1.2
	**32**	4.51 ± 0.85	0.01 ± 0.003	0.13 ± 0.04	0.97 ± 0.08	0.69 ± 0.08	0.77 ± 0.04	1.1
Reference	**Cisplatin**	15.28 ± 2.63	7.38 ± 1.53	18.25 ± 3.11	9.15 ± 2.07	2.22 ± 1.02	22.52 ± 3.15	10.10
	**Oxaliplatin**	1.15 ± 0.43	4.37 ± 1.07	8.25 ± 3.42	1.01 ± 0.34	1.53 ± 0.88	3.06 ± 1.00	2.00

**Table 16 ijms-22-03817-t016:** Antiproliferative activity of platinum(IV) prodrugs **27–32** on HEK293 (non-cancerous embryotic kidney) cell line. Selectivity index (SI) = IC_50_ (normal cell line)/IC_50_ (tumor cell line).

			Selectivity Index (SI)		
Class	Compound/Cell Line	IC_50_, µM HEK293	PSN-1	BCPAP	LoVo
PDKi, COXi	**27**	0.09 ± 0.02	1.5	1.5	0.1
	**28**	0.07 ± 0.03	0.9	0.5	0.2
HDACi, COXi	**29**	0.11 ± 0.01	1.2	1.8	2
	**30**	0.13 ± 0.03	0.1	1.6	0.6
	**31**	0.47 ± 0.08	6.7	0.7	13.8
	**32**	0.59 ± 0.21	4.5	59	0.6
Reference	**Cisplatin**	19.62 ± 2.33			

**Table 17 ijms-22-03817-t017:** Cellular uptake, DNA platination, and the ability of prodrugs **27–32** to inhibit HDAC and COX-2 and cause mitochondria depolarization in PSN-1 (pancreas cancer) cells.

Class	Compound	Cellular Uptake, ng Pt/10^6^ Cells	DNA platination, ng Pt/µg DNA	HDAC Inhibition, %	COX-2 Inhibition, %	Cells with Depleted Mitochondrial Potential, %
PDKi, COXi	**27**	5	0.3	<1	9	15
	**28**	20	1.5	<1	9	28
HDACi, COXi	**29**	13	0.8	8	6.1	39
	**30**	6	2.3	9	7.9	29
	**31**	17	0.75	7.5	11.2	32
	**32**	7.5	0.2	8	4.1	25
	**Cisplatin**	3	1.2	1	3	3

**Table 18 ijms-22-03817-t018:** Antiproliferative activity of prodrugs **33a, 33** and **34** on LNCaP (prostate cancer), DU145 (prostate cancer), MDA-MB-231 (triple negative breast adenocarcinoma), MCF-7 (breast cancer), and HCT-116 (colon cancer) cell lines. MCR-5 pd30 (fetal lung normal cell line) was used as a reference. Selectivity index (SI) = IC_50_ MCR-5 pd30/average IC_50_ (LNCaP and DU145).

Compound/ Cell Line	IC_50_, µM						
	LNCaP	DU145	MDA-MB-231	MCF-7	HCT116	MCR-5 pd30	SI
**33**	0.09 ± 0.02	0.15 ± 0.03	0.18 ± 0.01	0.45 ± 0.06	0.32 ± 0.08	2.2 ± 0.3	18.3
**33a**	0.24 ± 0.06	0.19 ± 0.04	0.54 ± 0.06	0.7 ± 0.1	0.9 ± 0.1	3.8 ± 0.6	17.7
**34**	0.26 ± 0.03	0.27 ± 0.01	0.6 ± 0.1	0.8 ± 0.1	1.0 ± 0.2	3.6 ± 0.2	13.6
**Estramustine**	6 ± 1	11 ± 1	5.5 ± 0.9	12.6 ± 0.9	26 ± 2	25 ± 3	2.9
**Cisplatin**	2.4 ± 0.2	21 ± 3	14.8 ± 0.9	9 ± 1	2.4 ± 0.2	9 ± 2	2.6

**Table 19 ijms-22-03817-t019:** Cellular uptake in LNCaP cells preincubated with 10 µM of prodrugs **33, 34** and cisplatin for 6 hours and their corresponding logP values.

Compound	Cellular Uptake, ng Pt/10^6^ Cells	LogP
**33**	172 ± 37	0.18 ± 0.04
**34**	139 ± 16	−0.01 ± 0.08
**Cisplatin**	3.7 ± 0.8	−2.3 ± 0.4

## Data Availability

Not applicable.

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
