# Peer review of "Pt(IV) Prodrugs with NSAIDs as Axial Ligands"

_ijms, 2021, doi:10.3390/ijms22083817_

Round 1

Reviewer 1 Report

I believe this paper is not suitable for the International Journal of Molecular Sciences. It would be more appropriate for Pharmaceutics/Pharmaceuticals or Molecules. There are several paragraphs that do not contain any references although the authors don't present their original data. Also, there are some scientific errors (CML is not breast cancer!) 

  1. "Lobaplatin is approved in China for the treatment of chronic myelogenous leukemia (CML), an inoperable metastatic breast cancer" - CML is not a metastatic breast cancer, but a blood cell disorder !!!
  2. Lines 99 - 132: where are the references?
  3. lines 140-150: references?
  4. lines 152-161: references? 
  5. etc

Author Response

Dear Editor, dear reviewers

Thank you for your thorough reading of this review and your comments. The list of answers/corrections can be found below:

Reviewer 1:

"Lobaplatin is approved in China for the treatment of chronic myelogenous leukemia (CML), an inoperable metastatic breast cancer" - CML is not a metastatic breast cancer, but a blood cell disorder !!!

Answer: we have corrected the phrasing of the sentence to reflect the fact that lobaplatin was approved for treatment of several cancers and disorders: “Lobaplatin is approved in China for the treatment of chronic myelogenous leukemia (CML), and inoperable metastatic breast cancer and small-cell lung cancer [3]”. The corresponding citation was added to evade confusion.

Lines 99 - 132: where are the references?

Answer: lines 101 – 116 are devoted to the studies on Platin-A metabolism and anti-inflammatory activity. In this part we reported the results of said studies and the conclusion the authors derived from the results obtained. In the end of this part a reference to the article by Pathak et al. was added to make the understanding of this part clearer. Lines 119 – 135 are devoted to the article by Neumann et al. and the results reported. In the end of this part the reference to the paper by Neumann et al was added as well.

lines 140-150: references?

Answer: lines 140 – 150 are devoted to article by Neumann et al. This part contains reference [23] (a reference to the synthetic procedure) on line 143. On line 145 there are references [30, 31] on influence of platinum(IV) prodrugs reduction potential on antiproliferative activity.

lines 152-161: references?

Answer: lines 151 – 161 are devoted to the results reported by Neumann et al. In this paragraph the results obtained and reported in the paper are discussed. The conclusions stated in the paragraph are made by the authors. The reference to the article by Neumann et al is added to the end of the paragraph to evade confusion.

Etc

Answer: the present review contains through analysis of research devoted to platinum(IV) prodrugs with NSAIDs as axial ligands. If the reviewer would like us to introduce additional references in the certain parts of the review, we would be glad to do so.

Reviewer 2 Report

In this article, the authors summarize the studies devoted to the development of Pt(IV) prodrugs with NSAIDs as axial ligands, the mechanism of their cytotoxic action, the anti-inflammatory activity, the structure-activity ratio, and therapeutic efficacy. The manuscript is straightforward, well written, and concise and has clear results within the scope of a review article. Definitely deserves to be published and is a valuable contribution to the “International Journal of Molecular Sciences”. Some minor flaws need to be addressed before publication.

Minor points:

[1] “Introduction”, Page 1/29, Lines 34-36:

Protein binding to cisplatin leads to the formation of toxic metabolites that cause serious side effects such as kidney damage, nerve damage, and hearing loss [5].”.

At that point, please report that a peak in urinary metabolites of serotonin occurs 6 hours after cisplatin administration, suggesting a strong correlation of serotonin release and vomiting with this agent. Delayed emesis occurs 24 hours or more after chemotherapy has been administered. Among chemotherapeutic agents, cisplatin causes the most severe delayed emesis.

Recommended reference: Boussios S, et al. Systemic treatment-induced gastrointestinal toxicity: incidence, clinical presentation and management. Ann Gastroenterol. 2012;25(2):106-118.

[2] 2.4.Naproxen”, Page 11/29, Lines 291-293:

The pro-apoptotic proteins BAD and BAX were upregulated in the presence of all three platinum compounds in both cell lines, while anti-apoptotic gene BCL-2 was downregulated.”.

There is evidence that the upregulation of the BAX increases the activity of the caspases and enhances the apoptotic activity. It would be beneficial for the readers to be mentioned at that point that many preclinical ovarian cancer studies correlate BCL-2-regulated apoptosis to metformin’s chemosensitizing effects. The chemosensitizing effect of metformin seems to be correlated with p53 function. Furthermore, metformin may re-sensitize platinum-resistant ovarian cancer cells to chemosensitive cells, either by induction of autophagy or via its anti-inflammatory properties.

Recommended reference: Boussios S, et al. Wise Management of Ovarian Cancer: On the Cutting Edge. J Pers Med. 2020;10(2):41.

[3] “3.2.Ibuprofen, Aspirin/PDK,PhB,Val, HDAC inhibitors”, Page 19/29, Lines 518-520:

Inhibition of HDAC was shown to lead to chromatin de-condensation, thus making cellular DNA more sensitive to platination [54]”.

Please, incorporate that there is also available preclinical evidence of synergies between HDAC-inhibitors and anti-PD-L1 immune checkpoint blockade in ARID1A-deficient ovarian cancer, which may be a promising combination to be evaluated in future clinical trials.

Recommended reference: Samartzis EP, et al. Endometriosis-associated ovarian carcinomas: insights into pathogenesis, diagnostics, and therapeutic targets-a narrative review. Ann Transl Med. 2020 Dec;8(24):1712.

Author Response

Dear Editor, dear reviewers

Thank you for your thorough reading of this review and your comments. The list of answers/corrections can be found below:

In this article, the authors summarize the studies devoted to the development of Pt(IV) prodrugs with NSAIDs as axial ligands, the mechanism of their cytotoxic action, the anti-inflammatory activity, the structure-activity ratio, and therapeutic efficacy. The manuscript is straightforward, well written, and concise and has clear results within the scope of a review article. Definitely deserves to be published and is a valuable contribution to the “International Journal of Molecular Sciences”. Some minor flaws need to be addressed before publication.

Minor points:

[1] “Introduction”, Page 1/29, Lines 34-36:

“Protein binding to cisplatin leads to the formation of toxic metabolites that cause serious side effects such as kidney damage, nerve damage, and hearing loss [5].”.

At that point, please report that a peak in urinary metabolites of serotonin occurs 6 hours after cisplatin administration, suggesting a strong correlation of serotonin release and vomiting with this agent. Delayed emesis occurs 24 hours or more after chemotherapy has been administered. Among chemotherapeutic agents, cisplatin causes the most severe delayed emesis.

Recommended reference: Boussios S, et al. Systemic treatment-induced gastrointestinal toxicity: incidence, clinical presentation and management. Ann Gastroenterol. 2012;25(2):106-118.

Answer: the emesis as the strong side effect of cisplatin therapy was added along with the corresponding reference

[2] “2.4.Naproxen”, Page 11/29, Lines 291-293:

“The pro-apoptotic proteins BAD and BAX were upregulated in the presence of all three platinum compounds in both cell lines, while anti-apoptotic gene BCL-2 was downregulated.”.

There is evidence that the upregulation of the BAX increases the activity of the caspases and enhances the apoptotic activity. It would be beneficial for the readers to be mentioned at that point that many preclinical ovarian cancer studies correlate BCL-2-regulated apoptosis to metformin’s chemosensitizing effects. The chemosensitizing effect of metformin seems to be correlated with p53 function. Furthermore, metformin may re-sensitize platinum-resistant ovarian cancer cells to chemosensitive cells, either by induction of autophagy or via its anti-inflammatory properties.

Recommended reference: Boussios S, et al. Wise Management of Ovarian Cancer: On the Cutting Edge. J Pers Med. 2020;10(2):41.

Answer: the correlation between BAX promoting apoptosis and caspase activity was added along with the corresponding reference.

[3] “3.2.Ibuprofen, Aspirin/PDK,PhB,Val, HDAC inhibitors”, Page 19/29, Lines 518-520:

“Inhibition of HDAC was shown to lead to chromatin de-condensation, thus making cellular DNA more sensitive to platination [54]”.

Please, incorporate that there is also available preclinical evidence of synergies between HDAC-inhibitors and anti-PD-L1 immune checkpoint blockade in ARID1A-deficient ovarian cancer, which may be a promising combination to be evaluated in future clinical trials.

Recommended reference: Samartzis EP, et al. Endometriosis-associated ovarian carcinomas: insights into pathogenesis, diagnostics, and therapeutic targets-a narrative review. Ann Transl Med. 2020 Dec;8(24):1712.

Answer: the suggested facts were added along with the corresponding article.

Round 2

Reviewer 1 Report

The authors have answered my comments. The authors should pay attention that each separate paragraph that does not contain their results or ideas is referenced. CML is not really treated anywhere around the world with platinum derivatives, the mainstay of treatment are tyrosine kinase inhibitors which have provided excellent results so far. Other hematological malignancies are however targeted with platinum derivatives, e.g., cisplatinum:

https://doi.org/10.1016/B978-0-12-817890-4.00006-8

Author Response

Dear Editor, dear reviewer,

Our answers to the reviewer comments could be found below:

The authors have answered my comments. The authors should pay attention that each separate paragraph that does not contain their results or ideas is referenced.

We have taken the remarks of the reviewer into consideration and made sure that the paragraphs contain references.

CML is not really treated anywhere around the world with platinum derivatives, the mainstay of treatment are tyrosine kinase inhibitors which have provided excellent results so far. Other hematological malignancies are however targeted with platinum derivatives, e.g., cisplatinum: https://doi.org/10.1016/B978-0-12-817890-4.00006-8

We have removed the CML from the text so the scope of the introduction would be limited only to oncologic diseases. The reference that contained the information on lobaplatin use in CML treatment was replaced with another one.
